# Adaptive Regret for Bandits Made Possible: Two Queries Suffice

**Zhou Lu**[*]
Princeton University

**Qiuyi Zhang**[*]
Google Deepmind

**Xinyi Chen**
Google Deepmind
Princeton University

**Fred Zhang**
UC Berkeley

**David P. Woodruff**
Google Research
Carnegie Mellon University[†]

**Elad Hazan**
Google Deepmind
Princeton University

## Abstract

Fast changing states or volatile environments pose a significant challenge to online optimization, which needs to perform rapid adaptation under limited observation. In this paper, we give query and regret optimal bandit algorithms under the strict notion of strongly adaptive regret, which measures the maximum regret over any contiguous interval $I$. Due to its worst-case nature, there is an almost-linear $\Omega(|I|^{1-\varepsilon})$ regret lower bound, when only one query per round is allowed [Daniely el al, ICML 2015]. Surprisingly, with just two queries per round, we give Strongly Adaptive Bandit Learner (StABL) that achieves $\widetilde{O}(\sqrt{n|I|})$ adaptive regret for multi-armed bandits with $n$ arms. The bound is tight and cannot be improved in general. Our algorithm leverages a multiplicative update scheme of varying stepsizes and a carefully chosen observation distribution to control the variance. Furthermore, we extend our results and provide optimal algorithms in the bandit convex optimization setting. Finally, we empirically demonstrate the superior performance of our algorithms under volatile environments and for downstream tasks, such as algorithm selection for hyperparameter optimization.

## 1 Introduction

In online optimization, a player iteratively chooses a point from a decision set, and receives loss from an adversarially chosen loss function. The classic metric for measuring the performance of the player is regret: the difference between her total loss and that of the best fixed comparator in hindsight.

However, as pointed out by Hazan and Seshadhri Hazan & Seshadhri (2009), regret incentivizes static behavior and is not the correct metric in changing environments. They instead proposed the notion of *adaptive regret*, defined as the maximum regret over any continuous interval in time. This notion intuitively captures adaptivity to the environment, and is studied later by Daniely et al. (2015) which gives an algorithm with near-optimal $O(\sqrt{|I|} \log T)$ adaptive regret for any interval $I$ simultaneously.

Most algorithms for minimizing adaptive regret, such as Hazan & Seshadhri (2009); Daniely et al. (2015), work by running $O(\log T)$ independent copies of online learning schemes and using a meta-learner to aggregate them. Therefore, they make $O(\log T)$ queries per round. In practice, oftentimes it is expensive to evaluate the arms. For example, an arm corresponds to a set of hyperparameter values of a large machine learning model, and playing an arm amounts to evaluating the model performance and fully training the model. This motivates the study of query-efficient adaptive regret algorithms. Recently, Lu and Hazan Lu & Hazan (2023) give an improved algorithm with only $O(\log \log T)$ queries per round. However, the known lower bound in Daniely et al. (2015) only shows that in the bandit setting, near-optimal adaptive regret is impossible with one query. This leaves the following question:

---

[*]Equal contribution

[†]Part of this work was done while visiting the Simons Insitute for the Theory of Computing

*Can we design algorithms with near-optimal adaptive regret using fewer queries?*

In this paper we give an affirmative answer, showing that two queries per round is enough to guarantee an $\widetilde{O}(\sqrt{|I|})$ adaptive regret bound. The key ingredient of our algorithm is using the additional query for exploration under a special distribution on arms, which serves to construct unbiased loss estimators with small variance for both experts and the meta-learner.

## 1.1 OUR RESULTS

We design adaptive regret minimization schemes with the power to query additional arms. In the adversarial multi-arm bandit (MAB) problem, the algorithm selects and plays one arm per round. In our setting the algorithm can pick additional arms to query *in parallel to* selecting which arm to play, then the loss values of both the queried and played arms will be revealed, but the algorithm only suffers the loss of the arm played. This is similar to, but still different from the multi-point feedback model Agarwal et al. (2010), which suffers the average loss of the arms. The query complexity counts the total arm observations received, including the one played.

First, we prove that two queries suffice for $\widetilde{O}(\sqrt{nI})$ adaptive regret in the MAB setting (we slightly abuse the notation $I$ to denote its length $|I|$). Our algorithm runs an EXP3-type meta-algorithm on top of black-box bandit learners $\mathcal{A}$ which are EXP3. It uses the additional query to perform exploration every round for constructing a bounded unbiased loss estimator. Then each bandit algorithm together with the meta-algorithm gets updated according to the loss estimator. Together with the lower bound in Daniely et al. (2015), our algorithm gives a tight characterization of the query efficiency of adaptive regret minimization. Such $O(\log T)$ to 2 improvement in the number of queries needed is significant for computational efficiency, for applications such as hyperparameter optimization in expensive settings.

Table 1: Adaptive regret bounds and query efficiency in the adversarial multi-armed bandits setting.

| Algorithm | Adaptive regret bound | Number of queries |
|---|---|---|
| FLH Hazan & Seshadhri (2009) | $\sqrt{nT}$ | $O(\log T)$ |
| SAOL Daniely et al. (2015) | $\sqrt{nI \log T}$ | $O(\log T)$ |
| EFLH Lu & Hazan (2023) | $I^{\frac{1}{2}+\varepsilon} \cdot \sqrt{n \log T}$ | $O\left(\frac{\log\log T}{\varepsilon}\right)$ |
| **This paper** (Theorem 1) | $\sqrt{nI \log n} \cdot \log^{1.5} T$ | 2 |

Next, we extend our MAB method to the bandit convex optimization (BCO) setting, proving that three queries suffice for $\widetilde{O}(\sqrt{I})$ adaptive regret. We use one of the additional queries to do uniform exploration, which gets us the loss value of some expert $i$'s prediction $\ell_t(\mathcal{A}_i(t))$. This value is used to construct both a bounded unbiased loss estimator to the meta EXP3 algorithm, and an unbiased sparse gradient estimator with bounded variance by leveraging another query to get $\ell_t(\mathcal{A}_i(t) + \mu)$, where $\mu$ is a small random perturbation term.

Finally, we show empirically the advantage of using our adaptive algorithms in changing environments on synthetic and downstream tasks, such as algorithm selection during hyperparameter optimization.

## 1.2 RELATED WORK

**Adaptive Regret Minimization**   Motivated by early work on shifting experts Herbster & Warmuth (1998); Bousquet & Warmuth (2002), the notion of adaptive regret was first proposed by Hazan & Seshadhri (2009). They gave an algorithm called Follow-the-Leading-History (FLH) with $O(\log^2 T)$ adaptive regret for online optimization on strongly convex functions. The bound on the adaptive regret for general convex cost functions, however, is $O(\sqrt{T} \log T)$. This question has been further studied by a sequence of work Daniely et al. (2015); Jun et al. (2017); Lu et al. (2022) which aims to improve the regret bound. In particular, Daniely et al. (2015) proposed the algorithm Strongly-Adaptive-Online-Learner (SAOL) achieving a near-optimal $O(\sqrt{I} \log T)$ bound. Later, Jun et al. (2017) improved this bound by a $\sqrt{\log T}$ factor and Lu et al. (2022) attained a second-order bound. Crucially, note that all the algorithms use $O(\log T)$ queries per round.

Recently, Lu & Hazan (2023) initiated a study on improving the efficiency of adaptive regret minimization. They showed a trade-off between adaptive regret and computational complexity, that an $\widetilde{O}(I^{\frac{1}{2}+\varepsilon})$ regret bound is achievable with $O\left(\frac{\log\log T}{\varepsilon}\right)$ base algorithms, each with one query per round. This result differs from ours in two aspects: (i) Lu & Hazan (2023) studied the computational complexity (number of base algorithms), while we consider the query complexity and achieve a better bound; and (ii) Lu & Hazan (2023) focused on improving the classic exponential-history-lookback technique of Hazan & Seshadhri (2009); Daniely et al. (2015) itself, by showing an regret upper bound with doubly exponential lookback. However, they also showed a matching lower bound that the history-lookback technique must use $\Omega(\log\log T)$ experts to achieve non-vacuous regret. Instead, our work bypasses such limitations by incorporating the EXP3 method into the classic framework, using only a constant number of queries.

For adaptive regret in the BCO setting, Zhao et al. (2021) showed that an $\widetilde{O}(\sqrt{T})$ adaptive regret bound is achievable under the two-point feedback model. However, the result is weaker than ours in two aspects. First, our result provides a strongly adaptive regret bound $\widetilde{O}(\sqrt{I})$ which depends on the interval-length, and is a stronger notion than adaptive regret. Second, our regret bound is independent of the condition number $\kappa$, while their regret bound is $\widetilde{O}(dGD\kappa\sqrt{T})$. We will discuss the comparison in more details later.

**Dynamic Regret** Dynamic regret is another related notation to handle changing environments in OCO, which aims to capture comparators with bounded total movement instead. Starting from Zinkevich (2003) which provided an $\widetilde{O}(\mathcal{P}\sqrt{T})$ dynamic regret bound of vanilla OGD where $\mathcal{P}$ denotes the path length, there have been many works on improving and applying dynamic regret bounds. Zhao et al. (2020); Zhang et al. (2017) achieved improved dependence on $\mathcal{P}$ under further assumptions. Baby & Wang (2021; 2022) focused on the setting of exp-concave and strongly-convex loss, showing that the classic adaptive regret algorithm FLH Hazan & Seshadhri (2009) guarantees $\widetilde{O}(T^{\frac{1}{3}}\mathcal{P}^{\frac{2}{3}})$ dynamic regret. Some works study the relationship between adaptive regret and dynamic regret, for example Zhang et al. (2018; 2020).

**Bandit Convex Optimization** The study on BCO was initiated by Flaxman et al. (2005), which showed how to construct an approximate gradient estimator with bandit feedback, providing an $O(T^{3/4})$ regret bound. If multi-point feedback is allowed, the regret bound can be improved to $O(\sqrt{T})$ as shown in Agarwal et al. (2010). Later, Bubeck et al. (2017) used a kernel-based algorithm to achieve the optimal $O(\sqrt{T})$ regret bound. Nevertheless, none of the prior works considered adaptive regret in the BCO setting.

## 2 SETTINGS AND PRELIMINARIES

We study adaptive regret in a limited information model, for both the multi-armed bandit (MAB) and the continuous bandit convex optimization (BCO) problems. In the MAB problem, a decision maker $\mathcal{A}$ plays a game of $T$ rounds, she pulls an arm $x_t$ at round $t$ and chooses an additional set of arms $X_t$ to query. Then an adversary reveals the values of the loss vector $\ell_t$ on $\{x_t\} \cup X_t$. The goal is to minimize the (strongly) adaptive regret, which examines all contiguous intervals $I = [j,s] \in [T]$: (we will omit $I$ when it's clear from the context)

$$\text{SA-regret}(\mathcal{A}, I) = \max_{s-j=I}\left[\sum_{t=j}^{s}\ell_t^\top e_{x_t} - \min_i\sum_{t=j}^{s}\ell_t^\top e_i\right].$$

For the BCO setting, similarly the decision maker chooses $x_t \in \mathcal{K} \subset \mathbb{R}^d$ at each time $t$ along with a set of points $X_t$, where $\mathcal{K}$ is some convex domain, then the adversary reveals the loss values on $\{x_t\} \cup X_t$. The player aims to minimize the (strongly) adaptive regret as well: (notice for expected regret, it contains the above definiton for MAB as a special case when $\mathcal{K}$ is a simplex)

$$\text{SA-regret}(\mathcal{A}, I) = \max_{s-j=I}\left[\sum_{t=j}^{s}\ell_t(x_t) - \min_{x\in\mathcal{K}}\sum_{t=j}^{s}\ell_t(x)\right].$$

The decision maker has limited power to query $\ell_t$. For every round $t$ she has a query budget $|X_t| \leq N$ for some small constant $N \in \mathbb{N}^+$. In particular, we will only consider the case $N = 1$ for the MAB problem and $N = 2$ for the BCO problem. We emphasize the choice of which arms to query is made before any information of the loss is revealed, thus differentiates our setting from the stronger "Bandit with Hints" setting Bhaskara et al. (2023). Our setting is stronger than the vanilla MAB setting, but weaker than the full-information setting or the hint setting.

We make the following assumption on the loss $\ell_t$ and domain $\mathcal{K}$, which is standard in literature.

**Assumption 1.** *For the MAB problem, the loss vector $\ell_t$ is assumed to have each of its coordinate bounded in $[0, 1]$. For the BCO problem, we assume the loss $\ell_t$ is convex, G-Lipschitz and non-negative. The convex domain $\mathcal{K}$ is sandwiched by two balls centered at origin: $r\mathbf{B} \subset \mathcal{K} \subset D\mathbf{B}$, we denote the condition number $\kappa = D/r$.*

## 2.1 THE QUERY MODEL

Our query model is similar to the multi-point feedback model Agarwal et al. (2010), with a slight difference. Both models specify a set of arms to query beforehand, then after the adversary reveals the loss, both models receive loss information on the set of chosen arms. Here, the definition (evaluation of the loss function at a given point) and complexity (number of evaluations) of query are the same for both models, and the only difference is that our model only incurs loss of the actually played arm, while the multi-point feedback model incurs the average loss of the set of chosen arms. Notice that neither model strictly implies the other model. As a result, it's fair to compare the query efficiency of our result with previous works under the multi-point feedback model.

## 2.2 THE EXP3 ALGORITHM

The EXP3 algorithm for adversarial MAB Auer et al. (2002) is based on the multiplicative update method, and performs a weight update according to an unbiased loss estimator using bandit feedback. Denote $w_t(k)$ as the weight of arm $k$ at time $t$, the EXP3 algorithm samples the arm $x_t$ according to the weights $w_t(k)$. Ideally, we would like to update each weight based on full information of the loss:

$$w_{t+1}(k) = w_t(k) \left( 1 - \eta \ell_t^\top e_k \right).$$

In the bandit setting, we have only the value of $\ell_t^\top e_{x_t}$. Let $\mathbf{1}$ denote the all-ones vector. The EXP3 algorithm instead constructs an unbiased estimator $\widehat{\ell}_t$ to replace $\ell_t$:

$$\widehat{\ell}_t(i) = \mathbf{1}_{i=x_t} \frac{\ell_t^\top e_{x_t}}{w_t(k)}.$$

This pseudo loss has a bounded variance. In particular, $\mathbb{E}[w_t^\top \widehat{\ell}_t^2] \leq n$, which gives the EXP3 algorithm an $\widetilde{O}(\sqrt{nT})$ regret bound.

## 2.3 STANDARD FRAMEWORK FOR MINIMIZING ADAPTIVE REGRET

We briefly review the standard framework of adaptive regret minimization. Since adaptive regret asks for small regret over all intervals, a simple idea is to construct, for any interval $I$, a base learner $\mathcal{A}_I$ that achieves optimal regret $O(\sqrt{I})$ on $I$. There are $O(T^2)$ contiguous intervals between $[1, T]$. Thus, a naive procedure is to run a meta-learner, such as multiplicative weights (MW) update, on these $O(T^2)$ experts, and this leads to an $O(\sqrt{I \log T})$ adaptive regret.

Unfortunately, although the regret bound is near optimal, the naive algorithm is inefficient. In particular, for any $t$ there are $\Omega(T)$ number of intervals containing $t$, and the naive algorithm needs to maintain and update $\Omega(T)$ number of base learners per round. To resolve this issue, the key technique is to consider only a subset $S$ of all intervals Hazan & Seshadri (2009); Daniely et al. (2015):

$$S = \left\{ [s2^k, (s+1)2^k - 1] \mid 0 \leq k \leq \log T, \ s \in \mathbb{N}^+, (s+1)2^k - 1 \leq T \right\}.$$

The set $S$, known as the geometric interval set, contains all successive intervals with length equal to some power of two. In particular, the $kth$ expert will only need to optimize the series of intervals $[2^k, 2 \times 2^k - 1], [2 \times 2^k, 3 \times 2^k - 1], \cdots$ one by one. With this technique, the algorithm only needs to hold a running set of base algorithms with size $O(\log T)$, while maintaining a near-optimal $\widetilde{O}(\sqrt{I})$ adaptive regret bound.

## 3 ADAPTIVE REGRET IN MULTI-ARMED BANDITS

In this section, we study efficient adaptive regret minimization in the adversarial multi-armed bandit (MAB) setting. We propose a procedure (Algorithm 1) that achieves $\widetilde{O}(\sqrt{nI})$ adaptive regret, using only two queries. At a high level, the algorithm maintains a set of instances of the EXP3 algorithm (i.e., base learners) and aggregates them via a MWU meta-algorithm.

---

**Algorithm 1** Strongly Adaptive Bandit Learner (StABL)

---

1: **Input:** general EXP3 algorithm $\mathcal{A}$ and horizon $T$.
2: Construct interval set $S = \{[s2^k, (s+1)2^k - 1] \mid 2 + \log \log T \leq k \leq \log T, s \in \mathbb{N}^+\}$.
3: Construct $B = \log T - (1 + \log \log T)$ independent instances of EXP3 algorithm $\mathcal{A}_k$, where $\mathcal{A}_k$ optimizes each $\{I \in S \mid 2^k = |I|\}$ one after another since they don't overlap.
4: Denote $w_t(k)$ to be the weight assigned to $\mathcal{A}_k$ at time $t$ by the meta-algorithm.
5: Denote $v(t, k) \in \mathbb{R}^n$ to be the distribution over arms by $\mathcal{A}_k$ at time $t$, and $v(t, k)_i$ to be the probability of sampling arm $i$ by $\mathcal{A}_k$ at time $t$.
6: Define $\eta_k = \min \left\{ 1/2\sqrt{n}, 1/\sqrt{n|2^k|} \right\}$, and initialize $w_1(k) = \eta_k$ for all $k \in [B]$.
7: **for** $\tau = 1, \ldots, T$ **do**
8:     Let $W_t = \sum_k w_t(k)$ and $p(t) = \frac{1}{W_t}(\ldots, w_t(k), \ldots)$ be the distribution over the base learners.
9:     For all $i \in [n]$, let

$$P(t)_i = \frac{\max_k v(t, k)_i^2}{2 \sum_i \max_k v(t, k)_i^2} + \frac{\sum_k v(t, k)_i}{2B}$$

    `// This defines a probability distribution over n arms.`
10:     Sample $x_t \sim \sum_k p(t)_k v(t, k)$, and in parallel sample $x_t' \sim P(t)$.
    `// Only the second sample x't will be used for weight updating.`

11:     Play $x_t$, suffer loss $\ell_t^\top e_{x_t}$ and observe loss $\ell_t^\top e_{x_t'}$. Compute loss estimator

$$\widehat{\ell}_t = \mathbf{1}_{i=x_t'} \frac{1}{P(t)_{x_t'}} \ell_t^\top e_{x_t'}.$$

12:     Update the weight $v(t+1, k)$ of each EXP3 instance with loss estimator $\widehat{\ell}_t$, via Algorithm 2.
13:     Update the meta-algorithm's weights over base learners via the loss estimator $\widehat{\ell}_t$. For each $k$, update $w_{t+1}(k)$ as follows,

$$w_{t+1}(k) = \begin{cases} \eta_k & 2^k \mid t+1 \\ w_t(k)\left(1 + \eta_k \widetilde{r}_t(k)\right) & \textbf{else} \end{cases}$$

    where $\widetilde{r}_t(k) = \widehat{\ell}_t^\top \sum_k p(t)_k v(t, k) - \widehat{\ell}_t^\top v(t, k)$.
14: **end for**

---

---

**Algorithm 2** Sub-routine: EXP3 with a General Loss Estimator

---

1: Input: horizon $T$, learning rate $\eta$ and $w_1 = \frac{1}{n}$.
2: **for** $t = 1, \ldots, T$ **do**
3:     Play $i_t \sim w_t$.
4:     Get unbiased estimator of loss $\widetilde{\ell}_t$.
5:     Update $y_{t+1}(i) = w_t(i)e^{-\eta \widetilde{\ell}_t(i)}$, $w_{t+1} = \frac{y_{t+1}}{\|y_{t+1}\|_1}$.
6: **end for**

---

**Theorem 1** (Adaptive regret minimization for multi-armed bandits). *For the multi-armed bandits problem with $n$ arms and $T$ rounds, Algorithm 1 achieves an expected adaptive regret bound of $O\left(\sqrt{nI \log n} \log^{1.5} T\right)$, using two queries per round.*

The main idea is to use EXP3-type algorithms for both the black-box base learners and the meta-algorithm in the typical adaptive regret framework. Directly using EXP3 in the MAB setting will fail, because the weight distribution might become unbalanced over time and the unbiased estimator of

the loss will have huge variance. When the unbiased estimator is propagated to the meta algorithm, it serves now as the value of "arm" which leads to a sub-optimal regret.

The remedy is to use an additional evaluation to create unbiased estimators of the loss vector with controllable variance, for updating both experts and the meta-algorithm. A naïve (but sub-optimal) choice is to sample uniformly over the arms to observe $x'_t$, and update both experts and the meta-algorithm (line 10 in Algorithm 1). By doing this, the loss estimator $\widehat{\ell}_t$ is not only unbiased, but also norm bounded by $n$. As a result, the variance term in the classical EXP3 analysis is bounded by $n^2$. Thus, the regret of the base learners, as well as the meta-algorithm, are $\sqrt{n}$ factor worse than optimal.

Instead of the naïve uniform exploration, we choose the distribution of the additional query to be the average of a uniform exploration among all base learners' choices and a near-optimal distribution for the meta-algorithm. Below we briefly explain the design of this importance sampling $P(t)_i$ (line 9 in Algorithm 1). $P_t$ is the average of two distributions, the first one controlling the regret of the meta learner while the second one controlling the regret of base learners. If $P_t$ is merely the second distribution, it achieves near-optimal regret for base learners, with the excessive term $\sqrt{n}$ improved to $\sqrt{\log T}$. However, the second distribution alone will make the regret of the meta learner unbounded, thus we mix it with the first distribution which is designed to control the regret of the meta learner. Such mixing is known to only affect the regret by a constant for EXP-3 type algorithms.

### 3.1 Proof Sketch

In the following, we slightly abuse notation and let $\ell_t^\top x = \ell_t^\top e_x$ for $x \in [n]$. The proof consists of three steps: decomposing the randomness of playing $x_t$, bounding the base learners' regret, and analyzing the regret of the meta-algorithm.

**Step 1:** There are two sources of randomness in the algorithm, namely, randomness **pl** in sampling which arm $x_t$ to play, and **ob** in sampling which arm $x'_t$ to observe and update weights. The key observation is that **pl** is independent of **ob**. Thus, we have the following equivalence via linearity and the tower property of expectations, for any fixed interval $I$ and arm $x^*$,

$$\mathbb{E}_{\mathbf{pl},\mathbf{ob}} \left[ \sum_{t \in I} \ell_t^\top x_t - \sum_{t \in I} \ell_t^\top x^* \right] = \mathbb{E}_{\mathbf{ob}} \left[ \sum_{t \in I} \widehat{\ell}_t^\top \sum_k p(t)_k v(t,k) - \sum_{t \in I} \widehat{\ell}_t^\top x^* \right],$$

which decouples the randomness of playing $x_t$ from observing $x'_t$.

**Step 2:** We prove the following lemma on EXP3 algorithms with general unbiased loss estimators, which gives a regret guarantee for Algorithm 2.

**Lemma 2** (Regret for EXP3). *Given $\widetilde{\ell}_t$, an unbiased estimator of $\ell_t$, such that for some distribution $z_t$, $\widetilde{\ell}_t(i) = \frac{1}{z_t(i)} \ell_t(i)$ and $\widetilde{\ell}_t(j) = 0$ for $j \neq i$ with probability $z_t(i)$. Suppose $z_t$ satisfies $w_t(i) \leq C z_t(i)$ for all $i$, where $w_t(i)$ represents the weight of the $i$th expert at time $t$. Algorithm 2 using $\widetilde{\ell}_t$ with $\eta = \sqrt{\frac{\log n}{TnC}}$ has regret bound $2\sqrt{CnT \log n}$.*

As a result, noticing $C = 2 \log T$ in our algorithm, we have that for any expert $\mathcal{A}_k$, its regret on interval $I$ can be bounded by $O(\sqrt{nI \log n \log T})$.

**Step 3:** We analyze the regret of the meta-learner, which is $\mathbb{E}_{\mathbf{pl, ob}} \left[ \sum_{t \in I} \ell_t^\top x_t - \sum_{t \in I} \ell_t^\top v(t,k) \right] = \mathbb{E}_{\mathbf{ob}} \left[ \sum_{t \in I} \widetilde{r}_t(k) \right]$. Define the pseudo-weight $\widetilde{w}_t(k)$ to be $\widetilde{w}_t(k) = \frac{w_t(k)}{\eta_k}$, and $\widetilde{W}_t = \sum_k \widetilde{w}_t(k)$, we first prove $\widetilde{W}_t \leq t(\log t + 1)$ using induction, which leads to the following estimate

$$\mathbb{E}_{\mathbf{ob}} \left[ \sum_{t \in I} \widetilde{r}_t(k) \right] \leq \eta_k \mathbb{E}_{\mathbf{ob}} \left[ \sum_{t \in I} \widetilde{r}_t^2(k) \right] + \frac{2 \log T}{\eta_k}.$$

Then we show the term $\mathbb{E}_{\mathbf{ob}} \left[ \widetilde{r}_t^2(k) \right]$ can be bounded by:

$$\mathbb{E}_{\mathbf{ob}} \left[ \widetilde{r}_t^2(k) \right] \leq \mathbb{E}_{\mathbf{ob}, <t} \left[ \sum_{i=1}^n \frac{(\ell_t^\top e_i)^2}{P(t)_i} \left( \max_k e_i^\top v(t,k) \right)^2 \right] \leq 2n \log T,$$

which implies the regret of our algorithm on any interval $I \in S$, can be bounded by $O(\log T \sqrt{nI \log n})$. Finally, such regret can be extended to any interval at the cost of an additional $\sqrt{\log T}$ term, by Cauchy-Schwarz (each arbitrary interval can be decomposed into a disjoint union of $O(\log T)$ geometric intervals as in Daniely et al. (2015)).

## 4 ADAPTIVE REGRET IN THE BCO SETTING

The result from the previous section inspires us to consider the following question: can we use a similar approach to achieve near-optimal adaptive regret in the bandit convex optimization (BCO) setting with constant number of queries? The answer is yes, that near-optimal adaptive regret is achievable with three queries, at the cost of extra logarithmic terms in the regret.

The algorithm for adaptive regret in the BCO setting (see appendix) has a similar spirit as Algorithm 1. At time $t$, each expert $k$ asks for gradient estimation at a point $\mathcal{A}_k(t)$, and we only randomly sample one expert to get the gradient estimation. If we sample expert $k_t$, the estimation for $\mathcal{A}_{k_t}$ is

$$\frac{dm\mathbf{u}(\ell_t(\mathcal{A}_{k_t}(t) + \delta_t \mathbf{u}) - \ell_t(\mathcal{A}_{k_t}(t)))}{\delta_t},$$

and the gradient estimation for the rest of experts is 0, where $m$ is the number of experts, $\delta$ is a small constant and $\mathbf{u}$ is a random unit vector. Such estimation is classic in the two-point feedback BCO setting Agarwal et al. (2010), in fact all we need is a (nearly) unbiased estimation of loss with bounded absolute value. Then we can use the same approach as the MAB case to make an unbiased estimation of the experts' losses, by setting

$$\widetilde{\ell}_t(\mathcal{A}_k(t)) = m\ell_t(\mathcal{A}_{k_t}(t))1_{[k=k_t]}.$$

We have the following regret guarantee for Algorithm 3. The proof is delayed to the appendix.

**Theorem 3.** *In the BCO setting, Algorithm 3 with $\delta_t = \frac{1}{\kappa T}$ uses three queries per round and achieves an expected adaptive regret bound of*

$$O\left(dGD\sqrt{I}\log^2 T\right).$$

Theorem 3 also directly applies to the full-information setting, improving the query complexity of all previous works in adaptive regret from $\Theta(\log T)$ to a constant, while preserving the optimal regret.

Compared to Zhao et al. (2021), note that though their algorithm uses one fewer query, they considered only the weaker notion of adaptive regret, failing to achieve dependence on interval length which leads to a worse $\sqrt{T}$ dependence on horizon. We discuss how to further reduce the required number of queries by one for strongly-adaptive regret in the appendix, by using the linear surrogate loss idea of Zhao et al. (2021) in our algorithm.

## 5 EXPERIMENTS

In this section, we evaluate the proposed algorithms on synthetic data and the downstream task of hyperparameter optimization. We note that even though the algorithms are stated in terms of losses, we find that in practice, using rewards instead of losses lead to better performance.

### 5.1 LEARNING FROM EXPERT ADVICE

**Experimental Setup** We first consider the learning with expert advice setting with linear rewards, and demonstrate the advantage of using an adaptive algorithm in changing environments. For simplicity, the arms and experts are equivalent in our construction. More specifically, there are $N$ arms, indexed from 0 to $N - 1$, and there are $N$ experts, where the $i$-th expert (zero-indexed) always suggests pulling the arm $i$. We take $N = 30$, time horizon $T = 4096$, and for each time step, we randomly generate a baseline reward $\widetilde{r}_t \in \mathbb{R}^N$ where $\widetilde{r}_{t,i}$ is drawn from the uniform distribution over $[0, 0.5]$. To simulate the volatile nature of the environment, we divide the time horizon into 4 intervals, and add an additional reward to the baseline reward of a different expert in each interval. In particular, for $t \in [0, 1023]$, we set the reward $r_{t,0} = \widetilde{r}_{t,0} + 0.5$ and $r_{t,i} = \widetilde{r}_{t,i}$ for $i \neq 0$; for $t \in [1024, 2047]$, we do the same for expert 1, and so on. The reward at each time step is $r_t^\top x_t$, where $x_t$ is the point played.

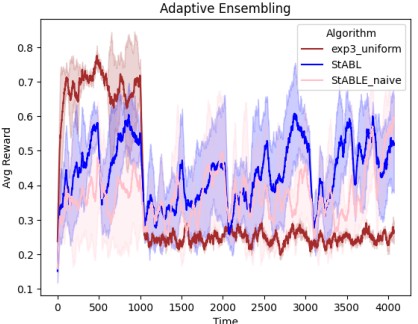 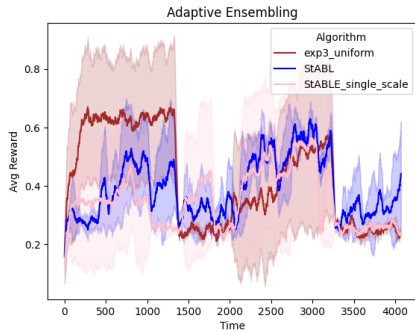

Figure 1: Comparison plots of the algorithm rewards in the learning with expert advice setting. The right subfigure shows the performance of the algorithms when the best arm changes at random intervals, and demonstrates the advantage of using base algorithms with varying history lengths

**Analysis**   In the first experiment, we compare the performance of the non-adaptive EXP3 algorithm with uniform exploration, StABL, and the StABL algorithm with naive exploration. In StABL with naive exploration, the observation query is sampled with the uniform distribution (Line 10 in Algorithm 1 is replaced with the uniform distribution). We run all algorithms 5 times and plot the moving average of their reward with a window of size 50 in the left subfigure of Figure 1. Both StABL and and StABL Naive can adapt to changing environments and recover after the best arm has changed. In contrast, EXP3 is optimal in the first interval, but its performance quickly degrades in the subsequent intervals. Compared to StABL, StABL Naive is worse in most intervals, reflecting the suboptimal performance resulting from the naive observation sampler.

To understand the importance of having base algorithms with varying history lengths, we also compare StABL with a variant, StABL Single Scale, where the base algorithm only has one history length of 1024. Such a variant would perform well in the previous experiment, since the intervals have the same length. In this experiment, we randomly generated time steps at which the best arm changes, and they are 1355, 1437, 1798, 3249, for a time horizon of 4096. As before, we run each algorithm 5 times and compare the moving average reward in the right subfigure of Figure 1. In this experiment, StABL Single Scale struggles in intervals that do not conform to a length of 1024. Thus, even though StABL requires more memory and compute by running $\log T$ base algorithms, it is more robust to volatile environments that can have irregular changes.

## 5.2   ALGORITHM SELECTION FOR HYPERPARAMETER OPTIMIZATION

**Algorithm as Arms**   We compare the effectiveness of bandit-feedback regret minimization algorithms for the downstream task of choosing the best algorithms for minimizing a blackbox function. Specifically, we view each round as a selection process between an ensemble of evolutionary strategies, each with different algorithm parameters, such as perturbation, gravity, visibility and pool size Yang & He (2013). Therefore each arm represents a specific evolutionary algorithm and the rewards of each arm are observed per-round in a bandit fashion, indicating the performance of the algorithm at that round, given all of the data observed so far.

**Benchmark and Reward Signal**   The reward is determined by the underlying task of blackbox optimization, which is done on Black-Box Optimization Benchmark (BBOB) functions (Tušar et al., 2016). BBOB functions are usually non-negative and we are in the simple regret setting, in which we want to find some $x$ such that $f(x)$ is minimized. Therefore we use the decrease of the objective, upon evaluation of the algorithm's suggestion, as the reward. Specifically, if $\mathcal{D}_i = (x_i, f(x_i))$ are all the evaluations so far in round $k$, then each algorithm $\mathcal{A}_j$ suggests $x_j = \mathcal{A}_j(\mathcal{D}_j)$ and the corresponding reward is $r_j = \max(\min_i(f(x_i)) - f(x_j), 0)$. Note that this reward is always positive and we normalize our BBOB functions so that it is within $[0, 1]$. Note that this reward signal is sparse, so we also add a regularization term of $-\lambda f(x_j)$ with $\lambda = 0.01$ of the negative objective into the reward. In addition to normalization, we also apply random vectorized shifts and rotations on these functions, as well as adding observation noise to the evaluations.

**Analysis**  We ran the four algorithms: the uniform random algorithm, classic EXP3 algorithm, the EXP3 algorithm with uniform exploration, and StABL with history lengths $[20, 40, 80, 160, 320, 640, 1280, 2560]$. We run each of our algorithms in dimensions $d = 32, 64$ and optimize for $1000, 2000$ iterations with $5$ repeats. For metrics, we use the log objective curve, as well as the performance profile score, a gold standard in optimization benchmarking (higher is better) Dolan & Moré (2002). Our full results are in the appendix and here, we focus on the SPHERE function (Fig. 2), where we see that StABL generally performs better adaptation than the EXP3 algorithms leading to better optimization throughout. Specifically, for the first half of optimization, both StABL and EXP3-Uniform enjoy a consistent performance advantage in performance, with respect to the Uniform strategy. However, this advantage sharply drops after around 600 Trials, after which the EXP3 strategy is no longer competitive against Uniform and ends up worse, as its performance curve abruptly stops at Trial 700. StABL generally tracks with the EXP3 strategy throughout this process but is noticeably better at adapting and maintaining its advantage after the critical turning point.

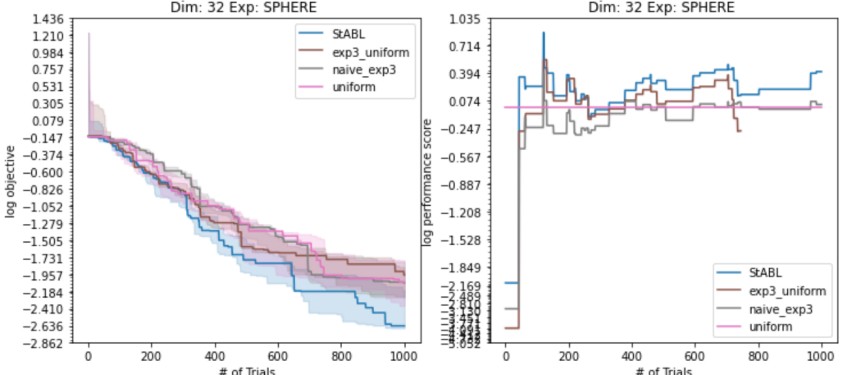

Figure 2: Algorithm comparison plots of the log objective (lower is better) and the performance profile score against the Uniform baseline (higher is better) for minimizing the 32-dimensional SPHERE across 1000 trials.

## 6   CONCLUSIONS

We study adaptive regret in the limited observation model. By making merely one additional query with a carefully chosen distribution, our algorithm achieves the optimal $\widetilde{O}(\sqrt{nI})$ adaptive regret bound in the multi-arm bandit setting. This result not only improves the state-of-the-art query efficiency of $O(\log \log T)$ in Lu & Hazan (2023), but matches the lower bound in the bandit setting Daniely et al. (2015), thus providing a sharp characterization of the query efficiency of adaptive regret. As an extension, we prove that the optimal $\widetilde{O}(\sqrt{I})$ adaptive regret can be achieved in the bandit convex optimization setting, with only two additional queries. We also conduct experiments to demonstrate the power of our algorithms under changing environments and for downstream tasks. We list some limitations of this work and potential directions for future research, elaborated below.

**Reduce the logarithmic dependence in $T$:** Although our regret bound is tight in the leading parameters $n, I$, it involves a worse dependence on $\log T$. For the full-information setting, the best known result in Orabona & Pál (2016) only contains an $\sqrt{\log T}$ dependence. The main open question is thus: can the $\log^{1.5} T$ factor be improved in general?

**Fewer queries for BCO:** Our BCO Algorithm uses two additional queries to achieve the optimal adaptive regret bound, however for the MAB setting Algorithm 1 requires only one query. Though BCO is a harder problem than MAB, and the lower bound of Daniely et al. (2015) was designed for MAB, we wonder whether an algorithm can use even fewer queries in the BCO setting.

**Extension to dynamic regret:** In the full-information setting, any advance in adaptive regret of exp-concave loss implies improved dynamic regret bounds via a black-box reduction, as shown in Lu & Hazan (2023). However, it seems harder in the bandit setting. The fundamental difficulty is, for the meta-learner, how to exploit exp-concavity (or strong convexity) of the loss in the MAB setting.

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
