## A   BCO ALGORITHMS

---

**Algorithm 3** Strongly Adaptive BCO Learner

---

1: **Input:** OCO algorithm $\mathcal{A}$ (Algorithm 4) and horizon $T$.
2: Construct interval set $S = \{[s2^k, (s+1)2^k - 1] \mid 2 + \log\log T \le k \le \log T, s \in \mathbb{N}^+\}$.
3: Construct $B = \log T - (1 + 2\log\log T)$ independent instances of the expert BCO algorithm $\mathcal{A}_k$, where $\mathcal{A}_k$ optimizes each $\{I \in S | 2^k = |I|\}$ one after another since they don't overlap.
4: Denote $w_t(k)$ to be the weight assigned to $\mathcal{A}_k$ at time $t$ by the meta algorithm.
5: Define $\eta_k = \frac{1}{GD}\min\left\{\frac{1}{2}, \sqrt{\frac{\log T}{2^k}}\right\}$, and initialize $w_1(k) = \eta_k$.
6: **for** $\tau = 1, \ldots, T$ **do**
7:    Let $W_t = \sum_k w_t(k)$, denote $p(t) = \frac{1}{W_t}(..., w_t(k), ...)$ as the distribution over experts.
8:    Denote $\mathcal{A}_k(t)$ to be the prediction of expert $\mathcal{A}_k$ at time $t$.
9:    Sample $k_t$ uniformly from $[B]$, sample a random unit vector $\mathbf{u}$ and choose constant $\delta_t$.
10:    Play $x_t = \sum_k p(t)_k \mathcal{A}_k(t)$ and suffer loss $\ell_t(x_t)$. Observe losses $\ell_t(\mathcal{A}_{k_t}(t) + \delta_t \mathbf{u})$ and $\ell_t(\mathcal{A}_{k_t}(t))$.
11:    Updating experts: construct gradient estimation for each $\mathcal{A}_k$ as below, then invoke Algorithm 4

$$\frac{d\log T\mathbf{u}(\ell_t(\mathcal{A}_{k_t}(t) + \delta_t\mathbf{u}) - \ell_t(\mathcal{A}_{k_t}(t)))}{\delta_t}\mathbf{1}_{[k=k_t]}$$

12:    Updating weights: construct loss vector estimations

$$\widetilde{\ell}_t(\mathcal{A}_k(t)) = \log T\ell_t(\mathcal{A}_{k_t}(t))\mathbf{1}_{[k=k_t]}, \ \widetilde{\ell}_t(x_t) = \sum_k p(t)_k\widetilde{\ell}_t(\mathcal{A}_k(t))$$

13:    Update the meta algorithm's weights. For each $k$, update $w_{t+1}(k)$ as follows,

$$w_{t+1}(k) = \left\{ \begin{array}{ll} \eta_k & 2^k | t+1 \\ w_t(k)(1 + \eta_k\widetilde{r}_t(k)) & \textbf{else} \end{array} \right.$$

   where $\widetilde{r}_t(k) = \widetilde{\ell}_t(x_t) - \widetilde{\ell}_t(\mathcal{A}_k(t))$.
14: **end for**

---

**Algorithm 4** Sub-routine: BCO with General Gradient Estimator

---

1: **Input**: horizon $T$, learning rate $\eta$ and $x_1 \in \mathcal{K}$.
2: **for** $t = 1, \ldots, T$ **do**
3:    Play $x_t$ and suffer loss $\ell_t(x_t)$.
4:    Get gradient estimator $\widetilde{g}_t$, such that there exists $a, b > 0$ and another proxy loss $\widehat{\ell}_t$, satisfying: $\mathbb{E}[\widetilde{g}_t] = \nabla\widehat{\ell}_t(x_t), \mathbb{E}[\|\widetilde{g}_t\|_2] \le b$ and $\forall x \in \widehat{\mathcal{K}}$ it holds $|\ell_t(x) - \widehat{\ell}_t(x)| \le a$.
5:    Update $x_{t+1} = \Pi_{\widehat{\mathcal{K}}}[x_t - \eta\widetilde{g}_t]$, where $\widehat{\mathcal{K}} = \{x | \frac{1}{1-\kappa\delta_t}x \in \mathcal{K}\}$ is the domain of $\widehat{\ell}_t$.
6: **end for**

---

## B   PROOF OF THEOREM 1

*Proof.* The proof consists of three steps: eliminating the randomness of playing $x_t$, analyzing the expert regret, and analyzing the meta algorithm regret.

**Step 1**

There are two sources of randomness in the algorithm, namely randomness in sampling which arm $x_t$ to play, and sampling which arm $x'_t$ to observe and update weights. Let **pl, ob** denote the randomness of selecting arms to play and to observe, respectively, over all iterations $T$, and let $\mathbb{E}$ denote the unconditional expectation.

The key observation here is that the randomness in playing is independent of the randomness in observing, affecting only the loss value suffered and nothing else. Based on this observation, the very first step of the proof is the following equivalence via linearity and tower property of expectation

$$
\begin{aligned}
\mathbb{E}\left[\sum_{t\in I}\ell_t^\top x_t - \sum_{t\in I}\ell_t^\top x^*\right] &= \sum_{t\in I}\mathbb{E}\left[\mathbb{E}\left[\ell_t^\top x_t - \ell_t^\top x^*|x_1,x'_1,\ldots,x_{t-1},x'_{t-1}\right]\right] \\
&= \sum_{t\in I}\mathbb{E}\left[\mathbb{E}_{\mathbf{ob}}\mathbb{E}_{\mathbf{pl}}\left[\ell_t^\top x_t - \ell_t^\top x^*|x_1,x'_1,\ldots,x_{t-1},x'_{t-1}\right]\right] \\
&= \sum_{t\in I}\mathbb{E}\left[\mathbb{E}_{\mathbf{ob}}\left[\ell_t^\top\sum_k p(t)_k v(t,k) - \ell_t^\top x^*|x_1,x'_1,\ldots,x_{t-1},x'_{t-1}\right]\right] \\
&= \sum_{t\in I}\mathbb{E}\left[\mathbb{E}_{\mathbf{ob}}\left[\widehat{\ell_t}^\top\sum_k p(t)_k v(t,k) - \widehat{\ell_t}^\top x^*|x_1,x'_1,\ldots,x_{t-1},x'_{t-1}\right]\right] \\
&= \sum_{t\in I}\mathbb{E}\left[\mathbb{E}_{\mathbf{ob}}\left[\widehat{\ell_t}^\top\sum_k p(t)_k v(t,k) - \widehat{\ell_t}^\top x^*|x'_1,\ldots,x'_{t-1}\right]\right] \\
&= \sum_{t\in I}\mathbb{E}_{\mathbf{ob}}\left[\widehat{\ell_t}^\top\sum_k p(t)_k v(t,k) - \widehat{\ell_t}^\top x^*\right] \\
&= \mathbb{E}_{\mathbf{ob}}\left[\sum_{t\in I}\widehat{\ell_t}^\top\sum_k p(t)_k v(t,k) - \sum_{t\in I}\widehat{\ell_t}^\top x^*\right],
\end{aligned}
$$

which decouples the randomness of playing $x_t$ from observing $x'_t$. The third equality holds as we take expectation over $x_t$ conditioned on $x_1, x'_1, \ldots, x_{t-1}, x'_{t-1}$, and the fourth equality holds since $\widehat{\ell_t}$ is an unbiased estimator of $\ell_t$. In the fifth equality, we remove the conditioning over $x_1, \ldots x_{t-1}$, since the random variables inside the conditional expectation are not functions of these variables. Hence, we can take expectation only over the randomness of the observations. As a result of this equivalence, the random $x_t$ term in the regret is replaced by the convex combination of $v(t, k)$, which is exactly the expectation of $x_t$ over the randomness of playing.

**Step 2**

Assume that the loss value $\ell_t(i)$ of each arm $i$ at any time $t$ is bounded in $[0, 1]$, to bound the regret of experts, we need the following lemma on EXP-3 algorithms with general unbiased loss estimators.

**Lemma 4.** *Given $\widetilde{\ell}_t$, an unbiased estimator of $\ell_t$, such that for some distribution $z_t$, $\widetilde{\ell}_t(i) = \frac{1}{z_t(i)}\ell_t(i)$ and $\widetilde{\ell}_t(j) = 0$ for $j \neq i$ with probability $z_t(i)$. Suppose $z_t$ satisfies $w_t(i) \leq C z_t(i)$ for all $i$. The EXP-3 algorithm using $\widetilde{\ell}_t$ with $\eta = \sqrt{\frac{\log n}{TnC}}$ has regret bound $2\sqrt{CnT\log n}$.*

*Proof.* Following the standard analysis of EXP-3 in Hazan (2016), we have that

$$\mathbb{E}[\text{regret}] = \mathbb{E}\left[\sum_{t=1}^{T} \ell_t(i_t) - \sum_{t=1}^{T} \ell_t(i^*)\right] \leq \mathbb{E}\left[\sum_{t=1}^{T} \widetilde{\ell}_t(w_t) - \sum_{t=1}^{T} \widetilde{\ell}_t(i^*)\right]$$

$$\leq \mathbb{E}\left[\eta \sum_{t=1}^{T} \sum_{i=1}^{n} \widetilde{\ell}_t(i)^2 w_t(i) + \frac{\log n}{\eta}\right]$$

$$= \eta \sum_{t=1}^{T} \mathbb{E}\left[\sum_{i=1}^{n} \widetilde{\ell}_t(i)^2 w_t(i)\right] + \frac{\log n}{\eta}.$$

Let $\mathbb{E}_t$ denote the expectation of a random variable conditioned on the randomness before time $t$ (before playing $i_t$). Notice that for each time step,

$$\mathbb{E}\left[\sum_{i=1}^{n} \widetilde{\ell}_t(i)^2 w_t(i)\right] = \mathbb{E}\left[\mathbb{E}_t\left[\sum_{i=1}^{n} \widetilde{\ell}_t(i)^2 w_t(i)\right] | t-1\right] = \mathbb{E}\left[\sum_{i=1}^{n} \ell_t(i)^2 \frac{w_t(i)}{z_t(i)}\right]$$

$$\leq \mathbb{E}\left[\sum_{i=1}^{n} \frac{w_t(i)}{z_t(i)}\right] \leq Cn$$

Therefore, taking $\eta = \sqrt{\log n / TCn}$, we have

$$\mathbb{E}[\text{regret}] \leq 2\sqrt{TCn \log n}.$$

$\square$

Back to our algorithm, had we used $v(t,k)$ to sample $x'_t$, then in the above lemma we have $C = 1$ for the $k$th expert, which implies an optimal regret $2\sqrt{nI \log n}$. However, each expert may have a very different $v(t,k)$, thus making one of them optimal can lead to worse regret on the rest of experts.

Instead, in our algorithm we use the distribution $P_t \geq \frac{1}{2B} \sum_k v(t,k)$ to sample $x'_t$, in which $\frac{1}{B} \sum_k v(t,k)$ is just the average of $v(t,k)$. For each expert $k$, this distribution treated as $z_t$ in the above lemma, guarantees $C = 2B \leq 2\log T$. Therefore, we have that for any expert $k$, its regret on interval $I$ is bounded by

$$\mathbb{E}[\text{regret of expert } k] = \mathbb{E}_{\mathbf{ob}}\left[\sum_{t \in I} \widehat{\ell}_t^\top v(t,k) - \sum_{t \in I} \widehat{\ell}_t^\top x^*\right] \leq 2\sqrt{2nI \log n \log T}.$$

**Step 3**

It's only left to check the regret of the meta algorithm. Our analysis follows Theorem 1 in Daniely et al. (2015) with a few changes. Recall the equivalence obtained in step 1, the term we need to bound is the regret of our action $x_t$ with respect to expert $k$ for all experts:

$$\mathbb{E}_{\mathbf{pl, ob}}\left[\sum_{t \in I} \ell_t^\top x_t - \sum_{t \in I} \ell_t^\top v(t,k)\right] = \mathbb{E}_{\mathbf{ob}}\left[\sum_{t \in I} \widetilde{r}_t(k)\right].$$

where we recall the definition $\widetilde{r}_t(k) = \widehat{\ell}_t^\top \sum_k p(t)_k v(t,k) - \widehat{\ell}_t^\top v(t,k)$.

Define the pseudo-weight $\widetilde{w}_t(k)$ to be $\widetilde{w}_t(k) = \frac{w_t(k)}{\eta_k}$, and $\widetilde{W}_t = \sum_k \widetilde{w}_t(k)$. We are going to prove $\widetilde{W}_t \leq t(\log t + 1)$. To this end, we fix any possible "trajectory" of $\widetilde{W}_t$'s that can be reached by our algorithm, then prove $\widetilde{W}_t$ is bounded using induction. Since the only source of randomness for the objective on the right hand side comes from the observations $x'_1, \ldots, x'_T$, we take an arbitrary sequence of $x'_t$ such that the weights of the algorithm are totally deterministic.

When $t = 1$, only $\mathcal{A}_1$ is active therefore $\widetilde{W}_1 = 1 \leq 1(\log 1 + 1)$. Assume now the claim holds for any $\tau \leq t$, we decompose $\widetilde{W}_{t+1}$ as

$$
\begin{aligned}
\widetilde{W}_{t+1} &= \sum_k \widetilde{w}_{t+1}(k) \\
&= \sum_{k, 2^k | t+1} \widetilde{w}_{t+1}(k) + \sum_{k, 2^k \nmid t+1} \widetilde{w}_{t+1}(k) \\
&\leq \log(t+1) + 1 + \sum_{k, 2^k \nmid t+1} \widetilde{w}_{t+1}(k),
\end{aligned}
$$

because there are at most $\log(t+1) + 1$ number of different intervals in $S$ starting at time $t+1$, where each such interval has initial weight $\widetilde{w}_{t+1}(k) = 1$. Now according to the induction hypothesis,

$$
\begin{aligned}
\sum_{k, 2^k \nmid t+1} \widetilde{w}_{t+1}(k) &= \sum_k \widetilde{w}_t(k)(1 + \eta_k \widetilde{r}_t(k)) \\
&= \widetilde{W}_t + \sum_k \widetilde{w}_t(k) \eta_k \widetilde{r}_t(k) \\
&\leq t(\log t + 1) + \sum_k w_t(k) \widetilde{r}_t(k).
\end{aligned}
$$

We complete the argument by showing that $\sum_k w_t(k) \widetilde{r}_t(k) = 0$. By the definition of $\widetilde{r}_t(k)$, we have that

$$
\begin{aligned}
\sum_k w_t(k) \widetilde{r}_t(k) &= W_t \sum_k p(t)_k \left( \widehat{\ell}_t^\top \left( \sum_k p(t)_k v(t,k) \right) - \widehat{\ell}_t^\top v(t,k) \right) \\
&= W_t \left( \widehat{\ell}_t^\top \left( \sum_k p(t)_k v(t,k) - \sum_k p(t)_k v(t,k) \right) \right) = 0.
\end{aligned}
$$

The second inequality holds because $\sum_k p(t)_k = 1$.

We notice that the weights are all non-negative. To see this, notice the non-negativity of $\widetilde{w}_t(k)$ is guarded by the non-negativity of $1 + \eta_k \widetilde{r}_t(k)$. Let's lower bound $\widetilde{r}_t(k)$: by the definition of $\widehat{\ell}_t$, it's a sparse vector where only the coordinate $x'_t$ can have a positive value $\frac{1}{P(t)_{x'_t}}$. However, $P(t)_{x'_t}$ is lower bounded by $\frac{v(t,k)_{x'_t}}{2B}$ by its definition (line 9), thus $\widetilde{r}_t(k) \geq -2B \geq -2\log T$. Since $k \geq 2 + \log \log T$, we have that $\eta_k \leq \frac{1}{4\log T}$, which ensures the desired non-negativity.

Because weights are all non-negative, we obtain

$$
t(\log t + 1) \geq \widetilde{W}_t \geq \widetilde{w}_t(k).
$$

Hence, using the inequality that $\log(1 + x) \geq x - x^2$ for $x \geq -\frac{1}{2}$, we have that for the $k$ satisfying $2^k = |I|$,

$$
\begin{aligned}
2\log t \geq \log(\widetilde{w}_t(k)) &= \sum_{t \in I} \log(1 + \eta_k \widetilde{r}_t(k)) \\
&\geq \sum_{t \in I} \eta_k \widetilde{r}_t(k) - \sum_{t \in I} (\eta_k \widetilde{r}_t(k))^2
\end{aligned}
$$

Rearranging the above inequality and taking an expectation over the observations, we get the following bound

$$
\mathbb{E}_{\mathbf{ob}} \left[ \sum_{t \in I} \widetilde{r}_t(k) \right] \leq \eta_k \mathbb{E}_{\mathbf{ob}} \left[ \sum_{t \in I} \widetilde{r}_t^2(k) \right] + \frac{2\log T}{\eta_k}.
$$

Next, we need to estimate $\widetilde{r}_t(k)$ with our observation, which we apply with the distribution $P(t)$. Denote $N_t = \sum_{i=1}^n \max_k v(t,k)_i^2$, the term $\mathbb{E}_{\mathbf{ob}}\left[\widetilde{r}_t^2(k)\right]$ can be bounded by:

$$\mathbb{E}_{\mathbf{ob}}\left[\widetilde{r}_t^2(k)\right] = \mathbb{E}_{\mathbf{ob},<t}\left[\mathbb{E}_{\mathbf{ob},t}\left[\left(\widehat{\ell}_t^\top \sum_k p(t)_k v(t,k) - \widehat{\ell}_t^\top v(t,k)\right)^2 |t-1\right]\right]$$

$$= \mathbb{E}_{\mathbf{ob},<t}\left[\sum_{i=1}^n P(t)_i \frac{(\ell_t^\top e_i)^2}{P^2(t)_i}\left(\left(\sum_k p(t)_k v(t,k)\right) - v(t,k)\right)_i^2\right]$$

$$\leq \mathbb{E}_{\mathbf{ob},<t}\left[\sum_{i=1}^n \frac{(\ell_t^\top e_i)^2}{P(t)_i}\left(\max_k e_i^\top v(t,k)\right)^2\right]$$

$$\leq \mathbb{E}_{\mathbf{ob},<t}\left[2nN_t\right].$$

where we use the fact that $\frac{(\ell_t^\top e_i)^2}{P(t)_i} \leq \frac{2N_t}{\max_k e_i^\top v(t,k)^2}$ by the definition of $P(t)$. We still need to upper bound the normalization term $N_t$. In fact, since $v(t,k)$ is a distribution for each $k$, we can bound the Frobenius norm by:

$$N_t \leq \sum_i \sum_k (e_i^\top v(t,k))^2 \leq \log T$$

because $\sum_i e_i^\top v(t,k) = 1$. As a result, we have that

$$\mathbb{E}_{\mathbf{ob}}\left[\sum_{t\in I} \widetilde{r}_t(k)\right] \leq 2\eta_k nI\log T + \frac{2\log T}{\eta_k} \leq 8\log T\sqrt{nI}.$$

Putting the two pieces together, we have that the regret of our algorithm on any interval $I \in S$ with length at least $4\log T$ (i.e. $k \geq 2 + \log\log T$), can be bounded by $O(\log T\sqrt{nI\log n})$. For other intervals with length at most $4\log T$, the $O\left(\sqrt{nI\log n}\log^{1.5} T\right)$ regret bound automatically holds (this is the reason we only hedge over $2 + \log\log T \leq k \leq \log T$ instead of $0 \leq k \leq \log T$).

Finally, apply the same argument as in A.2 of Daniely et al. (2015) (also Lemma 7 in Lu et al. (2022)). This allows us to extend the $O(\log T\sqrt{nI\log n})$ regret bound over $I \in S$ to any interval $I$ at the cost of an additional $\sqrt{\log T}$ term, by observing that any interval can be written as the union of at most $\log T$ number of disjoint intervals in $S$ and using Cauchy-Schwarz; see Theorem 1 in Daniely et al. (2015).

□

## C    PROOF OF THEOREM 3

*Proof.* Similar to the proof of Theorem 1, we first prove a regret bound of Algorithm 4. We follow the classic framework of "gradientless" gradient descent (see chapter 6.4 in Hazan (2016)). Let's focus on some expert $\mathcal{A}_k$ over an interval $[j,s]$ where $2^k|j,s$, the proxy loss $\widehat{\ell}_t$ for this expert is the smoothed version of $\ell_t$:

$$\widehat{\ell}_t(x) = \mathbb{E}_{\mathbf{u}\sim\mathbf{B}}\ell_t(x+\delta\mathbf{u})$$

We can verify that the properties in Algorithm 4 hold with constants $a = \delta G$ and $b = dG\log T$. As for the domain $\widehat{\mathcal{K}}$, we notice that (1) $\forall x \in \widehat{\mathcal{K}}, \mathbf{u}$ we have that $x + \delta\mathbf{u} \in \mathcal{K}$, therefore $\mathcal{A}_{k_t}(t) + \delta_t\mathbf{u}$ in Algorithm 1 is feasible; (2) for any point $x \in \mathcal{K}$, there exists another point $x' = \Pi_{\widehat{\mathcal{K}}}[x] \in \widehat{\mathcal{K}}$ such that $\|x - x'\|_2 \leq D\kappa\delta$ by the property of projection. These properties allow us to bound the regret of $\widehat{\ell}_t$ instead:

$$\sum_{t=j}^s \mathbb{E}[\ell_t(\mathcal{A}_k(t))] - \sum_{t=j}^s \ell_t(x) \leq \sum_{t=j}^s \mathbb{E}[\ell_t(\mathcal{A}_k(t))] - \sum_{t=j}^s \ell_t(x') + \kappa\delta DGT$$

$$\leq \sum_{t=j}^s \mathbb{E}[\widehat{\ell}_t(\mathcal{A}_k(t))] - \sum_{t=j}^s \widehat{\ell}_t(x') + (2+\kappa)\delta DGT.$$

As a result, we only need to analyze the regret bound of Algorithm 4 for the loss function $\widehat{\ell}_t$. By the standard analysis of OGD (see Theorem 3.1 in Hazan (2016)), we have that

$$\sum_{t=j}^{s} \mathbb{E}[\widehat{\ell}_t(\mathcal{A}_k(t))] - \sum_{t=j}^{s} \widehat{\ell}_t(x') \leq \eta \sum_{t=j}^{s} \mathbb{E}\left[\|\widetilde{g}_t\|_2^2\right] + \frac{D^2}{\eta}$$

$$\leq \eta I d^2 G^2 \log^2 T + \frac{D^2}{\eta}$$

Combining these two inequalities, we have that

$$\sum_{t=j}^{s} \mathbb{E}[\ell_t(\mathcal{A}_k(t))] - \sum_{t=j}^{s} \ell_t(x) \leq \eta I d^2 G^2 \log^2 T + \frac{D^2}{\eta} + (2+\kappa)\delta DGT = O\left(dGD\sqrt{I}\log T\right)$$

when we choose $\eta = \frac{D}{dG\sqrt{I}\log T}$ and $\delta = \frac{1}{\kappa T}$. This proves that each expert has a near-optimal expected regret bound over the intervals it focuses on.

Notice that it's crucial to look at the function value $\ell_t(\mathcal{A}_{k_t}(t))$ in Algorithm 3, otherwise $b$ will have an additional $\frac{1}{\delta}$ dependence which leads to sub-optimal regret bounds. This corresponds to the sharp separation between two-query Agarwal et al. (2010) and one-query Flaxman et al. (2005) settings.

The rest of the proof is almost identical to that of Theorem 1. Some details are even easier, since we only uniformly sample from the experts instead of using a complicated distribution as in Algorithm 1. The next step is to prove a regret bound of the meta MW algorithm. By a similar argument as step 1, for any interval $I = [j, s] \in S$, the actual regret over the optimal expert $\mathcal{A}_k$ on $I$ can be transferred as

$$\mathbb{E}\left[\sum_{t=j}^{s} \ell_t(x_t) - \sum_{t=j}^{s} \ell_t(\mathcal{A}_k)\right] = \mathbb{E}\left[\sum_{t=j}^{s} \widetilde{\ell}_t(x_t) - \sum_{t=j}^{s} \widetilde{\ell}_t(\mathcal{A}_k)\right],$$

which allows us to deal with the pseudo-loss $\widetilde{\ell}_t$ instead. We focus on the case that $\sqrt{\frac{\log T}{2^k}} \leq \frac{1}{2}$, because in the other case the length $I$ of the sub-interval is $O(\log T)$, and its regret is upper bounded by $IGD = O(GD\sqrt{\log T I})$, and the conclusion follows directly.

Define the pseudo-weight $\widetilde{w}_t(k)$ to be $\widetilde{w}_t(k) = \frac{w_t(k)}{\eta_k}$, and $\widetilde{W}_t = \sum_k \widetilde{w}_t(k)$. We are going to prove $\widetilde{W}_t \leq t(\log t + 1)$. When $t = 1$, only $\mathcal{A}_1$ is active therefore $\widetilde{W}_1 = 1 \leq 1(\log 1 + 1)$. Assume now the claim holds for any $\tau \leq t$, we decompose $\widetilde{W}_{t+1}$ as

$$\widetilde{W}_{t+1} = \sum_k \widetilde{w}_{t+1}(k)$$

$$= \sum_{k, 2^k | t+1} \widetilde{w}_{t+1}(k) + \sum_{k, 2^k \nmid t+1} \widetilde{w}_{t+1}(k)$$

$$\leq \log(t+1) + 1 + \sum_{k, 2^k \nmid t+1} \widetilde{w}_{t+1}(k),$$

because there are at most $\log(t+1) + 1$ number of different intervals in $S$ starting at time $t + 1$, where each such interval has initial weight $\widetilde{w}_{t+1}(k) = 1$. Now according to the induction hypothesis,

$$\sum_{k, 2^k \nmid t+1} \widetilde{w}_{t+1}(k) = \sum_k \widetilde{w}_t(k)(1 + \eta_k \widetilde{r}_t(k))$$

$$= \widetilde{W}_t + \sum_k \widetilde{w}_t(k)\eta_k \widetilde{r}_t(k)$$

$$\leq t(\log t + 1) + \sum_k w_t(k)\widetilde{r}_t(k).$$

We complete the argument by showing that $\sum_k w_t(k)\widetilde{r}_t(k) \leq 0$. By the definition of $\widetilde{r}_t(k)$ and convexity of $\widetilde{\ell}_t$, we have that

$$
\begin{aligned}
\sum_k w_t(k)\widetilde{r}_t(k) &= W_t \sum_k p(t)_k \left( \widetilde{\ell}_t \left( \sum_k p(t)_k \mathcal{A}_k(t) \right) - \widetilde{\ell}_t(\mathcal{A}_k(t)) \right) \\
&\leq W_t \sum_k p(t)_k \left( \sum_k p(t)_k \widetilde{\ell}_t\left(\mathcal{A}_k(t)\right) - \widetilde{\ell}_t(\mathcal{A}_k(t)) \right) \\
&= W_t \left( \sum_k p(t)_k \widetilde{\ell}_t\left(\mathcal{A}_k(t)\right) - \sum_k p(t)_k \widetilde{\ell}_t\left(\mathcal{A}_k(t)\right) \right) = 0.
\end{aligned}
$$

Hence, using the inequality that $\log(1+x) \geq x - x^2$ for $x \geq -\frac{1}{2}$, we have that for the $k$ satisfying $2^k = |I|$,

$$
\begin{aligned}
2\log t \geq \log(\widetilde{w}_t(k)) &= \sum_{t \in I} \log(1 + \eta_k \widetilde{r}_t(k)) \\
&\geq \sum_{t \in I} \eta_k \widetilde{r}_t(k) - \sum_{t \in I} (\eta_k \widetilde{r}_t(k))^2 \\
&\geq \eta_k \left( \sum_{t=j}^{s} \widetilde{r}_t(j, k) - \eta_k I G^2 D^2 \log^2 T \right),
\end{aligned}
$$

because $|\widetilde{r}_t(k)| \leq GD \log T$. Rearranging the above inequality and taking an expectation over the observations, we get the following bound

$$
\mathbb{E}\left[ \sum_{t \in I} \widetilde{r}_t(k) \right] \leq \eta_k I G^2 D^2 \log^2 T + \frac{2\log T}{\eta_k} = O(GD\sqrt{I}\log^{1.5} T).
$$

Combining this with the previous regret bound for experts, we have that for any such interval $I \in S$, the overall regret can be bounded by $O\left(dGD\sqrt{I}\log^{1.5} T\right)$. Finally, apply the same argument as in A.2 of Daniely et al. (2015) (also Lemma 7 in Lu et al. (2022)). This allows us to extend the regret bound over $I \in S$ to any interval $I$ at the cost of an additional $\sqrt{\log T}$ term, by observing that any interval can be written as the union of at most $\log T$ number of disjoint intervals in $S$ and using Cauchy-Schwarz; see Theorem 1 in Daniely et al. (2015). □

## D  AN ALTERNATIVE PROOF STRATEGY FOR BCO

As we mentioned in the main-text, it's possible to further improve the number of queries in our algorithm from three to two, by combining it with the linear surrogate loss idea from Zhao et al. (2021). We provide a brief algorithm description and proof here, based on the following observation. Let's denote $x_t$ as the point actually played by the algorithm and $g_t$ the sub-gradient of $\widehat{\ell}_t$ at $x_t$, then for any interval $I$ and any expert $\mathcal{A}_k$, we have that

$$
\text{convexity: } \sum_{t \in I} \widehat{\ell}_t(x_t) - \widehat{\ell}_t(x_I^{'*}) \leq \sum_{t \in I} g_t^\top (x_t - x_I^{'*}),
$$

$$
\text{decomposition: } \sum_{t \in I} g_t^\top (x_t - x_I^{'*}) = \sum_{t \in I} g_t^\top (x_t - \mathcal{A}_k(t)) + \sum_{t \in I} g_t^\top (\mathcal{A}_k(t) - x_I^{'*}).
$$

The above observation allows us to reduce to problem of minimizing the regret $\sum_{t \in I} \widehat{\ell}_t(x_t) - \widehat{\ell}_t(x_I^{'*})$, to minimizing the tracking regret $\sum_{t \in I} g_t^\top (x_t - \mathcal{A}_k(t))$ and the expert regret $\sum_{t \in I} g_t^\top (\mathcal{A}_k(t) - x_I^{'*})$ of an (adaptive) linear surrogate loss $h_t(x) = g_t^\top x$. Luckily, the linear loss is the same across all experts, thus we only need to make one gradient estimation of $g_t$ at each round.

The algorithm is similar to Algorithm 3, with a few differences:

1. in line 9, we don't sample $k_t$ anymore.

2. in line 10, we make only one additional observation $\ell_t(x_t + \delta_t \mathbf{u})$ instead.

3. in line 11, the gradient estimator is now constructed as $\widehat{g}_t = \frac{d\mathbf{u}(\ell_t(x_t + \delta_t \mathbf{u}) - \ell_t(x_t))}{\delta_t}$.

4. in line 12, $\widetilde{\ell}_t(\mathcal{A}_k(t)) = \widehat{g}_t^\top \mathcal{A}_k(t), \widetilde{\ell}_t(x_t) = \widehat{g}_t^\top x_t$.

Now we analyze the regret of the modified algorithm. The overall proof strategy is similar to that of Theorem 3. First we consider the regret of the pseudo loss $\widehat{h}_t(x) = \widehat{g}_t^\top x$. By the optimality of expert algorithms, we have that for some $k$ which corresponds to an expert $\mathcal{A}_k$ which optimizes $I$,

$$\sum_{t \in I} \widehat{g}_t^\top (\mathcal{A}_k(t) - x_I^*) = O(dGD\sqrt{I}),$$

because $\|\widehat{g}_t\|_2 \leq dG$ by definition. In addition, we have that

$$\sum_{t \in I} \widehat{g}_t^\top (x_t - \mathcal{A}_k(t)) = \widetilde{O}(GD\sqrt{I}),$$

by the same argument on the tracking regret as in the proof of Theorem 3. Use the fact that $\widehat{g}_t$ is an unbiased estimation of $g_t$, we reach the following upper bound on expected regret:

$$\mathbb{E}\left[\sum_{t \in I} \widehat{\ell}_t(x_t) - \widehat{\ell}_t(x_I^{'*})\right] \leq \mathbb{E}\left[\sum_{t \in I} g_t^\top (x_t - x_I^{'*})\right] = \mathbb{E}\left[\sum_{t \in I} \widehat{g}_t^\top (x_t - x_I^{'*})\right] = \widetilde{O}(dGD\sqrt{I}).$$

It's left to bound the estimation error between $\widehat{\ell}_t$ and the true loss $\ell_t$. As already shown in the proof of Theorem 3, similarly we have that

$$\sum_{t \in I} \mathbb{E}[\ell_t(x_t)] - \sum_{t \in I} \ell_t(x_I^*) \leq \sum_{t \in I} \mathbb{E}[\widehat{\ell}_t(x_t)] - \sum_{t \in I} \widehat{\ell}_t(x_I^{'*}) + (2 + \kappa)\delta DGT.$$

Combining the two upper bounds, the expected regret of this algorithm is bounded by $\widetilde{O}(dGD\sqrt{I} + \kappa\delta DGT) = \widetilde{O}(dGD\sqrt{I})$, with only two queries per round. To sum up, this algorithm combines the (1) strongly adaptive regret framework from our algorithm 3, (2) the linear surrogate loss idea from Zhao et al. (2021). (1) improves the sub-optimal $\widetilde{O}(\sqrt{T})$ adaptive regret of Zhao et al. (2021), while (2) reduces the required number of queries by one.

# E  ADDITIONAL EXPERIMENTS

We provide additional experiments in the learning from expert advice setting. The experimental setup is the same, but with more arms and time steps. In the additional experiments, we take $N = 300$, $T = 65536$, and we run all algorithms 10 times. We plot the moving average with a window of 500.

Our findings are largely consistent with the previous experiments. The left subfigure of Figure 3 corresponds to the performance of EXP3, StABL, and StABL Naive on four intervals of equal length, where in each interval we have a different best arm. It is clear that StABL attains the best rewards among the three algorithms. EXP3 has good rewards in the first interval, but is not able to adapt when the environment changes. In the right subfigure, we plot the performance of algorithms when the best arm changes at time 7853, 13822, 25180, and 56621. In this experiment, EXP3 underperforms both StABL and StABL Single Scale, while StABL Single Scale also has difficulty adapting to the different interval lengths, especially at the beginning where the environment changes quickly.

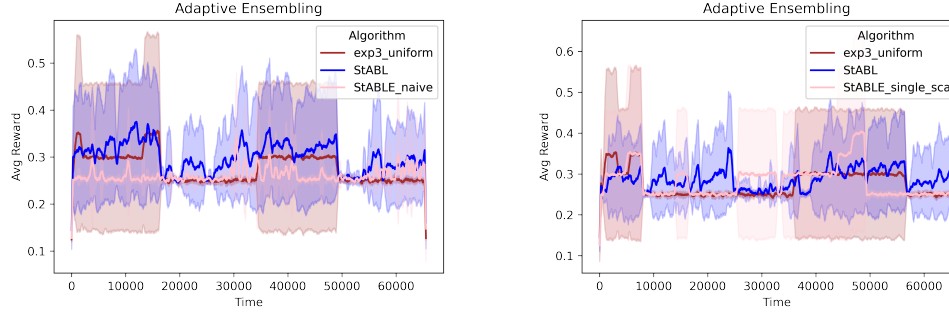

Figure 3: Further comparison plots of the algorithm rewards in the learning with expert advice setting.