# OpenReview forum: "Adaptive Regret for Bandits Made Possible: Two Queries Suffice"
_ICLR.cc/2024/Conference — ICLR 2024 poster_

### Official Review · Reviewer_EXm4 · 2023-10-26

**Soundness:** 3 good
**Presentation:** 2 fair
**Contribution:** 3 good
**Rating:** 8
**Confidence:** 4

**Summary:**

The paper investigated strongly adaptive regret under the multi armed bandit set up. In addition to standard multi arm bandit framework that the learner pulls a single arm and suffers a loss generated by the chosen arm, if the learner is allowed to query one extra arm at each iteration free of charge, then the proposed algorithm `StABL` achieves $O(\log T \sqrt{n I \log n} )$ strongly adaptive regret over any interval $I$ for a game consistent of $n$ expert span over a time horizon $T$.

In addition, a mirroring design of  `StABL`  can also be applied to solve bandit convex optimization problem if the learner has access to query two additional arms free-of-charge. This resultant to $O(dGD\sqrt{I} \log T )$ regret for $G$-Lipschitz loss and $d$-dimensional convex domain $\mathcal{K}$ that can be sandwiched between two $\ell_2$ ball with diameter $r, D$, respectively.

**Strengths:**

The paper is good at gathering key ingredients needed from previous literature, the intuition of the algorithm construction, and the necessity of the second query in order to achieve desired result.

The result is novel. Given previous lower bound on showing strongly adaptivity for bandit is impossible, this paper considers the nearly identical set-up to the impossible scenario and the presented result is optimal. Overall, in order to achieve the optimal regret in comparison to previous works which generally require $O(\log T)$ requires, the presented algorithm only require two queries for multi armed bandit set-up, three queries for the bandit convex optimization set-up. For the later case, it is also possible to achieve with two queries.

The experiment on synthetic data supports the construction of the algorithm adapts to changing environment. The proposed algorithm also excel in practical downstream tasks is demonstrated.

Overall, this paper is technical solid. The investigated problem is interesting, and the presented result is strong. It should be accepted provided the clarity of the write-up needs to be improved.

**Weaknesses:**

pg 4: sec2.2 second equation, LHS is a scalar, RHS is a vector

pg 4: sec 3. The abbreviation 'MWU' appeared the first time

In general, clarity of the write-up for section 3 needs to be improved.

Algo1: some part of the algorithm is ambiguous. For example, it might be more concise to gather all initialization parameters of exp3 base algorithms $\mathcal{A}_k$ together: ( Initialize $\mathcal{A}_k$ as algorithm 2 with learning rate ... and time horizon... ). Currently it is not immediately clear how $\mathcal{A}_k$ was initialized, and $v(t,k)$ is the $t^{th}$ output from $\mathcal{A}_k$ (a copy of algorithm 2) to the meta-algorithm. In addition, due to the preceding ambiguities, it causes confusion with $\eta_k$ defined in Algorithm 1 given Theorem 1 does not specify learning rate and time horizon for $\mathcal{A}_k$.

Algo 2: iterates are inconsistent, varying between $w$ and $x$. Also Lemma 4 is a general analysis for Algo2, it should quote algorithm 2 in stead of using `The EXP-3 algorithm`, which is even not consistent with the algorithm name itself.

Lemma 2: In the main text, given the context of Algorithm 2 $\mathcal{A}_k$, $w_t(i)$ is the probability for picking the $i^{th}$ arm. This is $v(t,k)_i$ defined in Algorithm 1. Not what was stated as the weight of the ith expert at time $t$. Due to this confusion, it is hard to see $C = 2\log T$. It only can be deduced after reading appendix at page 14.

It is also necessary to show the expressions of regret of expert k: $ \sum_{t \in I } \hat{ \ell}_t^T v(t,k) -  \hat{\ell}_t^T x^* $ in the main text instead of the appendix. Then it will become apparent why the quantity in Step 3 is needed, because `the regret of the Meta-learner` does not justify the quantity of interests in step 3.

Appendix B: pg15  first line. $\mathcal{A}_1$ by definition in Algorithm 1 line 3 optimizes interval with length of $2$. The index needs to be fixed.

Appendix C: paragraph starting after the first equation, 'Algorithm 3 is feasible'

**Questions:**

No

---

> ### Author Response · Authors · 2023-11-14
>
> Thank you for your valuable feedback!
>
> **Typos and ambiguous notations**: thanks for pointing them out! We will fix the typos and make the notations precise and consistent. In particular, we will add more details to explaining the algorithm.

---

### Official Review · Reviewer_gNKg · 2023-10-27

**Soundness:** 3 good
**Presentation:** 3 good
**Contribution:** 2 fair
**Rating:** 5
**Confidence:** 5

**Summary:**

This paper studies strongly adaptive online learning with bandits feedback.
Instead of making algorithms close to optimal over the whole time periods $1,\dots, T$, strongly adaptive algorithms ensures that the performance on every time interval is close to optimal.
This paper achieves $\mathcal{O}(\sqrt{n |I|})$ adaptive regret for multi armed bandits setting.
Especially, the proposed algorithms only require $2$ queries per round, which improves previous results requiring $O(\log T)$ queries.
The authors also extend their results to bandit convex optimization setting, and conduct experiments to show empirically the advantage of their algorithms.

**Strengths:**

- The idea is solid. Instead of reducing the number of base algorithms, the authors try to reduce the total number of interactions between the base algorithms and the environment. This setting makes sense when the cost of running base algorithms is less than the cost of interacting with the environment.
- Experimental results demonstrate the advantage of the design and show a remarkable improvement over the trivial EXP3 algorithm with uniform exploration.
- The paper is well writen. The proof in the appendix is well organized and mainly correct.

**Weaknesses:**

The main weaknesses is that there is essentially no difference between $2$ queries and $O(log T)$ queries: the high level idea of bounding the base learners regret is to use query $Uniform(A_1,\dots, A_B)$ instead of using $A_1,\dots, A_B$ independently.
Given that $B=\log T$, this method will at most scale the regret by O(log T) (as the results indeed show the regret scaling in this order), making it a trivial idea.
Both of $2$ queries and $O(log T)$ queries separate exploration and exploitation, which goes against the original intention of the bandits setting.

Besides, the authors should discuss the tradeoff of $O(log T)$ terms between regret and number of queries.
Note that in this problem the order of $O(log T)$ is not trivial: the number of base algorithms is only $O(log T)$.
Based on the current results, the cost of reducing the number of queries by $O(log T)$ times is an increase in regret by $O(log T)$ times (possibly more, see the Questions parts below).
This should be well specified in the paper.

**Questions:**

In the proof of Theorem 1, the author uses inequality $log(1+x)\ge x-x^2$ for $x\ge-1/2 (page 15).
Is there any proof to show $\eta_k \widetilde r_t(k)\ge -1/2$ for every $t$ and $k$?
Intuitively, $\widetilde r_t(k)\$ could be of order $v(t,k)_i/P(t)_i = O(log T)$, which implies that $\eta_k$ should be scaled $O(1/ log T)$.
This may leads to another $O(log T)$ term in the regret.

**Details Of Ethics Concerns:**

theory paper. No ethics converns.

---

> ### Author Response · Authors · 2023-11-14
>
> Thank you for your valuable feedback! Your comments are addressed below. We note that a log(T) to 2 improvement in the number of queries needed is significant for computational efficiency, for applications such as hyperparameter optimization in expensive settings.
>
> **Trade-off between regret and query**: We believe such trade-off is in fact non-trivial; however we will add a brief discussion. For example, as shown in Daniely et al 15, there is an $\tilde{O}(\sqrt{T})$ regret upper bound with $O(\log T)$ queries, and an $\Omega(T^{1-\epsilon})$ regret lower bound with one query. Here an $O(\log T)$ save in query actually costs an $\Omega(T^{\frac{1}{2}-\epsilon})$ term in regret, and it's non-trivial to show with two queries the cost in regret becomes logarithmic. In addition, a $\log T$ to $2$ improvement in the number of queries needed is significant for computational efficiency, for applications such as hyperparameter optimization in expensive settings.
>
> **Intention of the bandits setting**: the setting we consider is not the original bandit setting, for which a lower bound was already shown by Daniely et al 15. Instead, our main motivation is to reduce the number of queries when they are costly, which is naturally modeled by our multi-point feedback bandit-like setting.
>
> **Lower bound on $\tilde{r}_t(k)$**: thank you for pointing it out! This issue can be easily resolved, and we describe two ways below and will use the second method to clarify our paper.
>
> The straightforward way is by multiplying an $\frac{1}{2\log T}$ term to the choice of $\eta\_k$ in line 6 of Algorithm 1, at the cost of an additional $2\log T$ term in regret, which is negligible in our context. Let's lower bound $\tilde{r}\_t(k)$: by the definition of $\hat{\ell}\_t$ (line 11), it's a sparse vector where only the coordinate $x'\_t$ can have a positive value $\frac{1}{P(t)\_{x'\_t}}$. However, $P(t)\_{x'\_t}$ is lower bounded by $\frac{v(t,k)\_{x'\_t}}{2\log T}$ by its definition (line 9), thus $\tilde{r}\_t(k)\ge -2\log T$. Now, making $\eta\_k\le \frac{1}{4\log T}$ ensures the desired property that $\eta\_k \tilde{r}\_t(k)\ge-\frac{1}{2}$.
>
> The better way is by hedging over only the range $k\in [2+2\log \log T, \log T]$ instead of $[0,\log T]$ in the algorithm. To see this, by the definition of $\eta_k$, this issue doesn't exist when $2^k \ge 4\log T$, and the only problematic case is when we consider very small intervals with length $2^k< 4\log T$. However, for small intervals with length $O(\log T)$, our regret bound holds trivially and automatically, therefore this method will address the issue with no cost at regret.

---

> > ### Comment · Reviewer_gNKg · 2023-12-01
> >
> > Thanks for the clarifications. The rebuttal address most of my concerns.

---

### Official Review · Reviewer_kZqs · 2023-10-28

**Soundness:** 3 good
**Presentation:** 4 excellent
**Contribution:** 3 good
**Rating:** 8
**Confidence:** 3

**Summary:**

This paper shows that strongly adaptive regret is possible in bandit setting if one allows 2 queries in the MAB setting and 3 queries in the BCO setting. The regret is measured wrt only one of the queried points. In the MAB setting, the second query point is used as a free ticket for exploring the base learners: free ticket because, the authors don't care about the loss suffered by the exploration query.

**Strengths:**

- The authors uncover a previously unknown  phenomenon in the bandit setting.

- The presentation is clear.

- Experiments are conducted to validate theory.

**Weaknesses:**

- The main weakness (and hence the low score)  is due to a confusion I have regarding the paper. In the middle of page 15 in the supplementary material, the authors state that the weights are positive to get the inequality $\tilde W_t \ge \tilde w_t(k)$. Is the positivity proved somewhere in the paper? If not, can you provide the arguments for positivity of the weights?

I think this is a really interesting paper and I would be happy to recommend acceptance if the authors can clear this confusion.

- If we care about the average loss of both query points as in the multi-point feedback model of Agarwal et al 2010, does the pessimistic lowerbound on strongly adaptive regret still hold? A discussion on this can be helpful to readers. Also a discussion on the practical applications where your feedback model can be inadequate helps to understand the limitations of the work.

Other comments:
- In section 1.1, the authors say that their algorithm is run over black-box base bandit learners. Then in the description of Algorithm 1, they take the base learners to be EXP3, essentially not viewing the base learners as a black-box. This seems self-contradictory.

- Line 13, algorithm 1, unclear what $2^k | t+1$ means. I assumed that these are the inactive base learners

- Line 3 of algorithm 2, $w_{t}$ must be $x_{t}$

**Questions:**

see above

---

> ### Author Response · Authors · 2023-11-14
>
> Thank you for your valuable feedback! Your comments are addressed below.
>
> **Non-negativity of $\tilde{w}\_t(k)$**: thank you for pointing it out! This issue can be easily resolved, and we describe two ways below and will use the second method to clarify our paper.
>
> The straightforward way is by multiplying an $\frac{1}{2\log T}$ term to the choice of $\eta\_k$ in line 6 of Algorithm 1, at the cost of an additional $2\log T$ term in regret, which is negligible in our context. To see this, notice the non-negativity of $\tilde{w}\_t(k)$ is guarded by the non-negativity of $1+\eta\_k \tilde{r}\_t(k)$. Let's lower bound $\tilde{r}\_t(k)$: by the definition of $\hat{\ell}\_t$ (line 11), it's a sparse vector where only the coordinate $x'\_t$ can have a positive value $\frac{1}{P(t)\_{x'\_t}}$. However, $P(t)\_{x'\_t}$ is lower bounded by $\frac{v(t,k)\_{x'\_t}}{2\log T}$ by its definition (line 9), thus $\tilde{r}\_t(k)\ge -2\log T$. Now, making $\eta\_k\le \frac{1}{4\log T}$ ensures the desired non-negativity.
>
> The better way is by hedging over only the range $k\in [2+2\log \log T, \log T]$ instead of $[0,\log T]$ in the algorithm. To see this, by the definition of $\eta_k$, this issue doesn't exist when $2^k \ge 4\log T$, and the only problematic case is when we consider very small intervals with length $2^k< 4\log T$. However, for small intervals with length $O(\log T)$, our regret bound holds trivially and automatically, therefore this method will address the issue with no cost at regret.
>
> **Pessimistic lower bound**: Note that to get our bound, we essentially pick the "better" of both query points since we get a query for free. Therefore, when the algorithm incurs any $\Omega(1)$ fraction of the cost of the "worst" of both queries, the lower bound would therefore hold. Since the average loss would incur half the cost of the best and worst queries, it follows that the regret of the lower bound will appear.
>
> **Black-box experts**: by black-box we mean the expert can be any online optimization algorithm as long as it has the guarantee of Lemma 2, and EXP-3 is one of them. We will clarify this point.
>
> **Meaning of $2^k|t+1$**: here it means when $k$ satisfies $t+1$ is divisible by $2^k$, we restart the $kth$ expert by re-initializing its weight to be $\eta_k$.
>
> **$w_t$ must be $x_t$**: we will fix this typo.

---

> > ### Comment · Reviewer_kZqs · 2023-11-21
> > **Reply to authors**
> >
> > Thanks for the clarifications!

---

### Official Review · Reviewer_wKMs · 2023-11-01

**Soundness:** 3 good
**Presentation:** 3 good
**Contribution:** 3 good
**Rating:** 8
**Confidence:** 3

**Summary:**

The authors show strongly adaptive regret bounds for the algorithmic problems of prediction with experts and online convex optimization in a limited information setting. They specifically consider two problem settings:

1. A generalization of adversarial MABs (decision space is $\mathcal{K} = [n]$ set of arms) where, at each step $t \in [T]$, the learning algorithm chooses an arm $x_t \in [n]$ and suffers loss $\ell_t(x_t)$, but also is allowed to query the losses for an additional set of arms $X_t \subset [n]$ with $|X_t| \leq N$. $x_t$ and  $X_t$ are chosen in parallel, and the adversary chooses the losses after seeing $x_t, X_t$. So the learner gets the $(N+1)$-dim loss vector $[\ell_t(x) : x \in X_t \cup \\{x_t\\}]$ while suffering only the loss $\ell_t(x_t)$.
2. A generalization of Bandit Convex Optimization (BCO) where the regret is against points from a well-conditioned convex decision space $\mathcal{K} \subset \mathbb{R}^d$, where the learner again chooses $x_t \in \mathcal{K}$ and suffers the corresponding loss $\ell_t(x_t)$, but receives the loss values $[\ell_t(x) : x \in X_t \cup \\{x_t\\}]$ for some set $X_t$ with $|X_t| \leq N$.

In both settings, they bound the SA regret: $\text{SA-Regret}(\mathcal{A}, I) = \max_{[r,s] : s - r = I} \left[\sum_{t=r}^{s} \ell_t(x_t) - \min_{x \in \mathcal{K}} \sum_{t=r}^{s} \ell_t(x)\right]$

Their query-feedback model is very similar to multi-point bandit feedback (Agarwal et al, 2010), but with the crucial difference that the learner suffers loss $\ell_t(x_t)$ for the chosen arm rather than the average of the losses for all the queried arms.

They then show two main theoretical results:
1. A $2$-query algorithm ($N = 1$) for MABs with SA regret bound of $O(\sqrt{n I \log n} \cdot \log^{1.5} T) = \tilde{O}(\sqrt{nI})$.
2. A $3$-query algorithm ($N = 2$) for BCO with SA regret bound of $O(d G D \sqrt{I} \log^2 T) = \tilde{O}(\sqrt{I})$, where $\ell_t$s are $G$-Lipschitz and $\mathcal{K}$ is contained in the radius $D$ $\ell_2$-ball centred at the origin.

They also have basic experimental results with the implementations of the algorithm in synthetic, changing environments.

References
----------------
(Agarwal et al, 2010) Optimal Algorithms for Online Convex Optimization with Multi-Point Bandit Feedback.
(Daniely et al 2015) Strongly Adaptive Online Learning

**Strengths:**

* For MABs, the query complexity per round (2 queries) is optimal for getting sublinear adaptive regret (lower bound from (Daniely 2015)). It is also a significant improvement on earlier work (Lu and Hazan 2023), from $O(\log \log T)$ to $2$. In the regret bound, the dependence on $I$ is tight, whereas the dependence on $n$ is, I believe, tight up to a $\sqrt{\log n}$ factor. The query complexity is exactly optimal (for sublinear SA regret), and a significant improvement on earlier work.
* For BCO, both the query complexity and the regret together give substantial improvements on earlier work (query complexity improved from $\log T$ to $3$ for the existing $\tilde{O}(\sqrt{T})$-regret full-information algorithms, substantial improvement in the dependence of regret on $\sqrt{T}$ (weak) versus $\sqrt{I}$ (strong) when compared to the $2$-query adaptive regret algorithm of (Zhao et al 2021)).
* The algorithms innovatively combine existing techniques --- such as the geometric interval sets used by (Hazan and Seshadhri, 2009) and later work, as well as the SAOL-meta-learning approach used by (Daniely et al, 2015) --- with the Exp3 framework.

**Weaknesses:**

* The experimental results are a bit weak. For MABs, the $N (= 30)$ and $T (=4096)$ values are too small, and each algorithm is run only $5$ times to average out the reward curves. The experiments also compare their algorithm (StABL) with handicapped variants (StABL Naive and StABL Single Scale, which do illustrate certain requirements on the structure of StABL) as well as Exp3 (which is of course expected to perform badly in a changing environment). It would be interesting to see the practical tradeoff in the reward curve vs. computational efficiency, for say StABL and Full-EFLH from (Lu and Hazan 2023).

**Questions:**

(i) In page 3, paragraph 6, I believe that the "special case when $\mathcal{K}$ is a simplex" is a bit stronger than MAB (since it would effectively consider all convex combinations of arms).

(ii) In the description/pseudocode of Algorithm I, calling Algorithm 2 as a subroutine seems slightly ambiguous, since Algorithm 2 is written as a full learning loop (whereas the actual operation seems to be an update $v(t+1,k) \leftarrow v(t,k)$ for each $k$).

(iii) In page 6, end of section 3, "can be extended to any interval .... by Cauchy-Schwarz" seems very opaque. I would suggest at least mentioning that each arbitrary interval can be decomposed into a disjoint union of $O(\log T)$ geometric intervals as in (Daniely, 2015).

(iv) Including the code as supplementary material/link in the final version would enhance the value of having the experiments.

---

> ### Author Response · Authors · 2023-11-14
>
> Thank you for your valuable feedback! Your questions are addressed below.
>
> **Weak experiments**: We will add experiments with higher time horizon and repeats, as well as looking at the practical tradeoff in the reward curve vs. computational/sample efficiency. Note that the theory there for optimal rates are non-trivial but in practice, we can use the naive lower-variance estimator.
>
> **Question 1**: if we consider expected regret in MAB, it's equivalent to regret in simplex. We will clarify this point more carefully.
>
> **Question 2**: we will make it clear the calling of Algorithm 2 only concerns the weight updating.
>
> **Question 3**: we will add more details as you suggested.
>
> **Question 4**: Thanks for the suggestion! The code is now already implemented as part of an open source package. We will add a link to that OSS package for the camera-ready version, to avoid anonymity issues.

---

> > ### Author Response · Authors · 2023-11-23
> >
> > In Appendix E of the supplementary materials, we have expanded our experiments by increasing the number of arms $N$ to 300, and extending the time horizon $T$ to 65536. In addition, we averaged the results over 10 runs of each algorithm. The additional experimental results are largely consistent with our previous findings. However, due to time constraints, we were not able to complete the comparison with the full-information algorithm, given by Lu and Hazan.  Exploring the tradeoff between rewards vs. computational/sample efficiency would be an interesting topic for future investigations.

---

### Meta-Review · Area_Chair_YGWx · 2023-12-02

**Metareview:**

**Summary:**

The paper presents a significant advancement in strongly adaptive online learning with bandit feedback. It introduces algorithms that achieve optimal strongly adaptive regret in multi-armed bandit (MAB) and bandit convex optimization (BCO) settings with a small number of queries per round. In the MAB setting, the proposed algorithm, StABL, requires only two queries per round to achieve the desired regret, while in the BCO setting, three queries are sufficient. This approach substantially improves upon previous works that required a higher number of queries, demonstrating the paper's innovative contribution to reducing interaction costs in bandit settings.

**Strengths:**

1. The paper successfully addresses a previously challenging area in bandit settings, showing that strongly adaptive regret is achievable with a reduced number of queries.
2. It demonstrates optimal query efficiency, particularly in the multi-armed bandit setting, requiring only two queries per round, which is a substantial improvement over previous works.
3. The paper includes experiments on synthetic data that demonstrate the algorithm's adaptability to changing environments and its practical applicability in tasks like hyperparameter optimization.

**Weaknesses:**
1. For bandit convex optimization, the paper proposes a method requiring three queries. However, it's unclear if this number is optimal or necessary.
2. The paper lacks a discussion on the tradeoff between regret and the number of queries, an important aspect in bandit algorithms.
3. While the authors have successfully addressed a critical issue in the proof, as highlighted by one of the reviewers, ensuring such technical robustness throughout the paper is essential.

**Justification For Why Not Higher Score:**

The main limitations of the paper are the lack of exploration into the necessity of three queries for bandit convex optimization and the absence of a detailed analysis on the tradeoff between regret and the number of queries.

**Justification For Why Not Lower Score:**

Overall, the paper is recommended for acceptance, considering its significant contributions to the field, the novelty of the approach, and the authors' responsiveness in addressing the reviewers' concerns.

---

### Decision · Program_Chairs · 2024-01-16

Accept (poster)